# The Swine IFN System in Viral Infections: Major Advances and Translational Prospects

**DOI:** 10.3390/pathogens11020175

**Published:** 2022-01-27

**Authors:** Elisabetta Razzuoli, Federico Armando, Livia De Paolis, Malgorzata Ciurkiewicz, Massimo Amadori

**Affiliations:** 1National Reference Center of Veterinary and Comparative Oncology (CEROVEC), Istituto Zooprofilattico Sperimentale del Piemonte, Liguria e Valle D’Aosta, Piazza Borgo Pila 39/24, 16129 Genoa, Italy; livia.depaolis@izsto.it; 2Department of Pathology, University of Veterinary Medicine Hannover, Bünteweg 17, 30559 Hannover, Germany; federico.armando@tiho-hannover.de (F.A.); malgorzata.ciurkiewicz@tiho-hannover.de (M.C.); 3National Network of Veterinary Immunology (RNIV), Via Istria 3, 25125 Brescia, Italy; m_amadori@fastwebnet.it

**Keywords:** pig, interferons, viral infections

## Abstract

Interferons (IFNs) are a family of cytokines that play a pivotal role in orchestrating the innate immune response during viral infections, thus representing the first line of defense in the host. After binding to their respective receptors, they are able to elicit a plethora of biological activities, by initiating signaling cascades which lead to the transcription of genes involved in antiviral, anti-inflammatory, immunomodulatory and antitumoral effector mechanisms. In hindsight, it is not surprising that viruses have evolved multiple IFN escape strategies toward efficient replication in the host. Hence, in order to achieve insight into preventive and treatment strategies, it is essential to explore the mechanisms underlying the IFN response to viral infections and the constraints thereof. Accordingly, this review is focused on three RNA and three DNA viruses of major importance in the swine farming sector, aiming to provide essential data as to how the IFN system modulates the antiviral immune response, and is affected by diverse, virus-driven, immune escape mechanisms.

## 1. Introduction

Interferons (IFNs) are a family of cytokines that elicit pleiotropic biological effects and can be synthesized and secreted by most cell types. These proteins are characterized by antiviral activity, discovered in 1957 by Isaacs and Lindenmann [1] during studies on virus interference. They are considered the first line of defense initiated by cells during viral infections; on the whole, they show three major biological activities including antiviral, antitumor and immunomodulatory effects [2,3,4].

These properties are induced through the synthesis of many signaling proteins in a row. By binding to their respective receptors (Figure 1), they activate a network of distinct but related signaling pathways, such as the JAK-STAT pathway and the related signaling cascades. This results in the transcriptional regulation of several genes that promote the different pleiotropic responses elicited by IFNs [3,5], underlying a transient resistance to a broad range of viruses [2]. To date, three distinct classes of IFNs are known, i.e., Type I, Type II and Type III IFNs. Type II consists of one molecular species (IFN-γ) only, whereas Type I IFNs are a heterogeneous group characterized by several families (IFN-α, IFN-β, IFN-ε, IFN-ω, IFN-k, IFN-δ and IFN-τ). Moreover, some families, such as IFN-α, consist of different subtypes. Finally, type III IFNs include three molecular species (IL-29, IL-28A and IL-28B) with similar biological properties [6,7,8]. In pigs, within Type I IFNs, porcine IFN-β is a glycoprotein encoded by one gene, whereas IFN-α includes 17 genes [9] located on chromosome 1 [10]; these genes show 96–99.8% and 91.1–100% of nucleotide and amino acid identity, respectively [11]. In particular, Type I IFNs are produced by many cell types, in which they have a pivotal role in both direct and indirect innate immune responses. The biological effects of IFNs are mediated through IFN-regulated genes. Three major IFN-regulated pathways involving RNA-dependent protein kinase (PKR)/the eukaryotic initiation factor (eIF)-2a system, 2-5A synthetase/RNase L system and Janus Kinase (JAK)/signal transducer and activator of transcription (STAT) system have been identified [12]. In particular, the protein synthesis factor eIF-2a functions as the final effector of the PKR-induced pathway [12].

The IFN type I response to viral agents is often activated through a cGAS-STING DNA sensing pathway: after sensing and binding to DNA in the cytoplasm, the cyclic GMP-AMP synthase (cGAS) catalyzes the synthesis of GMP-AMP cyclic dinucleotide (cGAMP), which through a stimulator of interferon gene encoded protein (STING) signaling cascade leads to the recruitment of nuclear factor-kappa B (NF-κB) and interferon regulatory gene 3 (IRF 3) [13,14]. These transcription factors, once phosphorylated, are translocated to the nucleus and initiate the transcription of NF-κB and IRF3 [15,16].

By binding to their respective receptors, type I IFNs (α/β) are able to activate, through a JAK/STAT-mediated pathway, the transcription of several interferon stimulated genes (ISGs) which encode for several innate immune effectors [17].

The IFN system plays a pivotal role in the innate immune system and in the regulation of the adaptive immune response. However, to establish infection in host cells, viruses have evolved systems to escape IFN responses. For instance, during Classical Swine Fever Virus (CSFV) infection, type I IFN responses are inhibited by the degradation of Interferon Response Factor (IRF)3 [17]; Foot-and-Mouth Disease Virus (FMDV) produces L proteinase able to shut off protein synthesis [18]; Papillomavirus E6 and E7 oncoproteins can modulate both IFN type I production and its pathway [19,20,21].

Herpes simplex virus-1 (HSV-1) core protein activates SOCS-3, thereby down-regulating STAT and JAK [22], while hepatitis C virus (HCV) NS3/4A protease has the ability to inhibit RIG-I signaling and induce the cleavage of adaptor protein TRIF, thereby also disrupting TLR3 signaling. In many negative-strand RNA viruses, e.g., Rabies virus andEbola virus, an important viral polymerase complex, the phosphoprotein P, is the main IFN antagonist. This protein is able to interfere with the activation of IRF3 [23,24].

Thus, different viruses can inhibit the IFN system by several means, whichcan be referred to three distinct strategies: (1) Viral or virus-induced proteins are able to target components of the IFN response; (2) A single viral protein can block several stages of the IFN induction and signaling cascades; (3) A virus can induce diverse antagonistic activities, targeting different pathways of the IFN response. In addition to that, it is worth recalling that viruses can induce many proinflammatory cytokines and chemokines downstream of type 1 IFN-α/β receptor (IFNAR) signaling, which may cause an exaggerated inflammatory response and tissue injury. Thus, IFN-α production, in humans, may even be correlated with infection and illness severity. Moreover, it was shown that severe infection is associated with TRAIL-mediated apoptosis in both mice and humans, and that IFN-α/β can induce TRAIL expression by Influenza virus-infected macrophages and pDCs [25]. 

Following phosphorylation, STATs complexes translocate into the nucleus, where they bind IFN-stimulated response elements (ISRE) or gamma-activated sequences (GAS). This determines the transcription of hundreds of genes involved in antiviral response, including ISGs, IFNs, IRFs, STATs and NOS2.

In this review, we focus on the IFN response modulated by three RNA and three DNA viruses, sustaining diseases associated to severe morbidity and heavy economic losses in swine farms.

## 2. IFN Responses to RNA Viruses

In this section, we will expose the peculiarities of the observed IFN responses to infections sustained by three RNA viruses of major importance for farmed pigs.

### 2.1. Porcine Reproductive and Respiratory Syndrome

Porcine Reproductive and Respiratory Syndrome (PRRS) is still of major importance worldwide [26], the syndrome nowadays being enzootic in most countries within pig farms. Heavy direct and indirect losses are related to the common clinical signs of PRRS, i.e., late-term abortion in sows, mortality of piglets, respiratory disease, anorexia, cutaneous hyperemia in weaners and growers, as well as to the related Porcine Respiratory Disease Complex [27]. The two porcine Arteriviruses sustaining PRRS (PRRSV-1 and PRRSV-2) are still widely known as European (EU) type I, with the first strain isolated in 1991 and named “Lelystad”, and the North American (NA) type II, isolated in 1992 with the acronym ATCC VR-2332 [28]. It is worth stressing that the two viruses are currently identified as β Arterivirussuid 1 and β Arterivirussuid 2 by the International Committee on Taxonomy of Viruses [29]. As mentioned in our previous review [30], the clinical outcome of PRRSV infection is affected by three major components: virus virulence, host susceptibility and environmental stressors. In the framework of our topic, we will focus on the role of the IFN response in the context of PRRSV virulence.

#### 2.1.1. Innate Immune Responses to PRRSV

The dubious, inconclusive findings obtained in several studies on the immune response to PRRSV [31] question the very nature of the host/virus relationship. This can be largely referred to the impact of viral proteins on crucial checkpoints of the innate immune response, including the different branches of the IFN system [32]. In particular, at least three non-structural proteins (Nsp1, Nsp2 andNsp11) and the N nucleocapsid protein play an important role in the suppression of the IFN and NF-κB pathways [32]. As a result, PRRSV infection often leads to untimely, non-effective innate and adaptive immunity [31], including poor induction of cell-mediated immune responses [33,34]. Timely and effective innate immune responses are probably of major importance for the control of PRRSV infection, since early up-regulation of IL-1, IL-8 and IFN-γ genes in lymphoid tissues is correlated with successful virus clearance [35].

#### 2.1.2. IFN Sensitivity of PRRSV

PRRSV was shown to be sensitive to IFN-α both in vitro and in vivo [36]. In particular, recombinant porcine IFN-α can inhibit the growth of the virus in porcine alveolar macrophages (PAMs) [37]. The sensitivity of PRRSV to recombinant swine interferon beta (rswIFNβ) was clearly demonstrated in a previous study [38]; additionally, MARC-145 simian cells and porcine PAMs were shown to display different rswIFNβ-associated antiviral pathways [38].

IFN-γ profoundly affected PRRSV replication in porcine macrophages in terms of reduction in titer and percentage of positive cells, the effect being both dose-dependent and related to the time of exposure, despite unabated virus attachment [39].

Studies conducted in vitro by infecting monocyte-derived macrophages (MDMs) with PRRSV isolates of different in vivo pathogenicity showed that PRRSV sensitivity to type I and II IFN could differ depending on the virulence of the strain; in particular whileLVP and PR11 strains proved to be sensitive to the action of the IFN-β and IFN-γ, Lena, BOR59 and PR40 strains were found to be relatively resistant [40,41].

IFN-λ3 can significantly reduce the replication of PRRSV in PAMs, and such inhibition is dose- and time-dependent; plaque formation can be abrogated entirely, and virus yield is dramatically reduced when PAMs are treated with IFN-λ3 at 1000 ng/mL; these effects are in agreement with the expression of interferon-stimulated genes 15 (ISG15), 2’-5’-oligoadenylate synthase 1 (OAS1), IFN-inducible transmembrane 3 (IFITM3) and myxoma resistance protein 1(Mx1) in primary PAMs [42].

#### 2.1.3. Type I IFN Responses

PRRSV induces in vitro little if any synthesis of Type I IFNs in PAMs and monocyte-derived dendritic cells (MoDC); yet, a systemic IFN-αresponse was detected in vivo following infection with various PRRSV strains [30]. On the contrary, respiratory PRRSV infection is often characterized by poor local induction of IFN-α [37], as opposed to swine Influenza virus or Porcine Respiratory Coronavirus [43]. This is probably related to the aforementioned decoy strategies of the virus [31,32,33,34]. Taken together, the resulting picture is quite complex. Some in vitro findings point at a role of Type I IFNs in the control of PRRSV infection, as shown by the aforementioned viral strategies aimed at suppression of Type I IFN responses [30]. The role of Type I IFNs was also confirmed in studies on some MicroRNAs (miRNAs); thus, miR-26a inhibits and miR-373 promotes the replication of PRRSV by up- and down-regulating Type I IFN genes, respectively [44,45]. miR-382-5p [46] is up-regulated in PRRSV infection, with a consequent inhibition of polyI:C-induced Type I IFN production. On the other hand, the early interferon (IFN)-alpha response in vivo was reported as unfavorable marker in PRRSV-infected sows [47]. This is in line with our experience [48]: an attenuated and a virulent PRRSV strain induced early IFN-γ and IFN-α responses in weaners, respectively. This finding is not unusual; it is probably an example of the so-called “Bad IFN-α response”, widely known in the Influenza model [49], and also observed after Classical Swine Fever virus infection [50,51]. Tissue damage, immunopathology andcell death underlie the impact of unproper or mistimed Type I IFN responses [49].

The “Bad IFN-α Response” is probably related to a further, unfavorable marker: the late IL-10 response of PRRSV-infected pigs, as shown in our previous studies in vivo [52,53]. Such a response is probably accounted for by an important role of plasmacytoid dendritic cells (pDCs). These cells can release high-titered IFN-α following exposure to many PRRSV strains [36]. IFN-α can then induce IL-10 in LPS-stimulated monocytes and CD4+ T cells [54]. Interestingly, there is definitely a synergism in vivo between PRRSV and LPS [55], and an IL-10 response to some PRRSV strains was repeatedly observed in vitro by PRRS-naïve swine PBMC [56,57]. Last but not least, IFN-α and IL-10 can promote the differentiation of Type I T regulatory (Treg) cells [58]. Treg cells are also induced by PRRSV-infected dendritic cells [59], and this may further underlie virus-driven immunosuppression.

Type I IFN responses to PRRSV are likely to be severely inhibited when infection takes place through the antibody-dependent enhancement (ADE) pathway; following ADE, PRRSV is able to suppress the transcription of key antiviral genes such astumor necrosis factor-α (TNF-α) and interferon-β (IFN-β), which also implies the down-regulation of the genes coding for crucial transcription factors: interferon regulatory factor-1 (IRF-1), interferon regulatory factor-3 (IRF-3) and nuclear factor kappa B (NF-κB) [60].

#### 2.1.4. Type II IFN Responses

In swine infected by PRRSV, cell-mediated immune (CMI) responses are characterized by cytokine secreting cells (SC), mainly IFN-γ-secreting, CD8+ and CD4+/CD8+ double-positive T cells, detectable 2–3 weeks post-infection and showing an erratic behavior [61]. Such responses are usually detected in vitro by ELISPOT assays using tissue culture-adapted PRRSV as stimulating agent [62,63]. The correlation between IFN-γ SC and protection of sows is dubious [64]; a confounding factor may actually be the non-specific IFN-γ responses to stress antigens expressed in the established cell lines in which PRRSV is propagated [65]. With a proper control of such non-specific responses, PRRSV-specific IFN-γ response was transient and limited in our experience after PRRSV infection [66]. Viremic pigs do not show IFN-γ responses, which may be detected instead in suckling piglets [66], as a possible response of maternally-derived immune cells [67]. Natural Killer (NK) cells give rise to early IFN-γ responses to PRRSV infection [68], which check IL-10 production and down-regulate CD163 in macrophages [69], as well as their susceptibility to PRRSV infection [69]. The presence of CD3-, CD8+, allegedly NK cells in the endometrium of PRRSV-infected sows [70] also points at an important role of these cells. Interestingly, many PRRSV-viremic pigs show a large reduction of NK cell-mediated cytotoxicity [71], which may represent a further important feature in the host/virus relationship.

#### 2.1.5. Type III IFN Responses

Scanty data are available about the induction of Type III IFNs by PPRSV in vivo and in vitro. In a study on nsp2-deletion mutants of PRRSV propagated in PAM-pCD163 cells (a porcine monocyte-derived cell line), cytokine genes, including IL-8 andIL-10, were significantly elevated; in contrast, the expression of interferons (IFN-β, -γ and -λ) and antiviral genes was unchanged or down-regulated [72].

#### 2.1.6. Translational Prospects

Oral, low-dose IFN-α treatments during PRRS outbreaks proved clinically effective on farms in both sows and piglets [30]. These results are in agreement with the potent effector mechanisms of low-dose IFN-α shown in an in vitro model of pig tonsil cells [73], which highlights the oral lymphoid tissues as the crucial targets of this low-dose cytokine treatment. Reduced clinical signs were shown in pigs injected with a non-replicating human Adenovirus type 5 expressing porcine IFN-α and then challenged with PRRSV, in terms of lower febrile responses, decreased percentage of lung involvement and delayed viremia [74]. As for vaccines, porcine IFN-α was employed in the form of expression plasmids as vaccine adjuvant; this treatment led to significantly increased numbers of IFN-γ SC in PBMC after recall PRRSV stimulation in vitro; the same result was obtained with polyinosinic and polycytidilyc acid (Poly IC), a potent Type I IFN inducer in pigs [75].

On the whole, there is strong evidence of a high IFN sensitivity of PRRSV. This probably underlies the development of diverse andrelevant immune escape virus strategies. The outcome of these strategies is often a late, poor and erratic innate immune response, including IFNs and interferon-stimulated genes. Interestingly, a discrepancy can often be appreciated between clear systemic and little if any local Type I IFN responses; such a systemic response may even be an early unfavorable marker of PRRSV infection. The major role of IFN-γ SC is probably the modulation of macrophage permissiveness to PRRSV. The same kind of modulation is also exerted by mucosal IgA antibodies [76], which may underlie a useful synergism of activities toward effective disease control.

### 2.2. Foot-and-Mouth Disease

Foot-and-Mouth disease (FMD) is a highly contagious disease of cloven-hoofed animals including swine, and it is deemed as the economically most important animal disease; the disease was described for the first time in the 16th century and its main clinical, virological and immunological features have been extensively described for a long time. Clinical symptoms include fever, lameness, lymphopenia and vesicular lesions on the mouth, tongue, nose, feet and teats [77]. Swine are by far less susceptible than cattle to infection by the oral route, but can shed huge amounts of FMD virus (FMDV) as infectious aerosol, thus posing a serious threat to other farms [78]. The etiological agent is FMDV, an Aphthovirus of the family Picornaviridae; virions are non-enveloped and show icosahedral symmetry. Moreover, their genome consists of a single-stranded RNA of about 8500 nucleotides of positive polarity. Its replication takes place by means of a complementary, minus strand RNA. In this process, several errors accumulate so that FMDV is widely diverse, both genetically and antigenically [79]. This poses some hurdles for diagnosis and, most important, for the current FMD vaccination policies. In fact, the virus occurs as seven genetically and antigenically distinct serotypes: O, A, C, Asia 1 and Southern African Territories (SAT) 1–3; and within each serotype, multiple subtypes have been reported [80]. On the whole, disease control underlies annual costs from production losses and vaccination estimated at US$6.5-US$21 billion in FMD-endemic areas of the world [81].

#### 2.2.1. Innate Immune Response to FMDV

Many studies have addressed the crucial role of antibodies in the clearance of FMDV infection and in vaccine-induced protection. Until recently, limited research has been undertaken to investigate innate immune responses to FMDV [82]. Yet, some studies clearly demonstrated early protection after vaccination with little if any antibody production [83]. Emergency FMD vaccines induce in pigs diverseresponses, both Th1-like and Th2-like, without adverse reactions [84]. In particular, IL-6 and IL-8 did not relate to protection, whereas IL-12 production was highest in the protected vaccinated pigs; this points at a major role of monocytic cell activity for early protection in FMD-vaccinated pigs [85]. Highly potent (250–300 PD_50_) FMD vaccines developed in the Russian Federation were reported to confer protection in cattle as early as 1 day after injection, the protective effects being highest after vaccine administration in the mucous membranes of upper lips; although the authors provided evidence of FMDV receptor blockage in vitro, the involvement of an innate immune response is more credible in vivo to account for early protection [86]. In this respect, γδ T cells are probably pivotal to the early response to FMD vaccines. For instance, FMDV-naive porcine γδ T cells express in vivo mRNA of various cytokine and chemokine genes after injection of a high potency, emergency FMD vaccine [87]. A weak, ineffective innate immune response probably underlies the severity of FMD, since FMDV displays a plethora of immunosuppressive control actions aimed at checking the host’s innate immune response; the relevant effector mechanisms are based on both structural and non-structural viral proteins: FMDV VP1, VP3, L^pro^, 3C^pro^, 2B and3A can in fact affect pathways related to IFNs, NF-κB, eIF4G, histone H3 andcalcium balance, to name a few [88,89]. On the whole, FMDV can down-regulate the host’s innate immune response; such suppressive control actions take place at the initial stage of infection, thus enabling the virus to proliferate rapidly and spread to the natural sites of infection [89]. Among the different viral proteins involved in immunosuppression, the leader proteinase (L^pro^) probably plays a major role, since FMDV serotype A12 lacking the leader proteinase-coding region (A12-LLV2) does not induce signs of FMD in pigs [18].

As a result of a dynamic interaction, Type I IFNs are both induced and antagonized by the virus; induction of Type I IFNs follows the recognition of FMDV RNA by MDA5, and the released IFNs can curtail FMDV replication [90]. An acute phase response to FMDV infection has been described in cattle [91], whereas little if any data are known in pigs. Gross pathology and histopathological examinations confirm a serious inflammatory response to FMDV infection in pigs [78], but we still need to explore the relationship between the inflammatory response and clinical symptoms of FMDV infection [88].

Natural Killer (NK) cells could play a crucial role in the innate immune response to FMDV infection. NK cells in pigs are identified as CD2+/CD8+/CD3−cells; once they are activated by cytokines, such asIL-2, IL-12, IL-15, IL-18 or IFN-α, they increase their cytotoxicity against FMDV-infected target cells [82]. Yet, NK cells isolated from FMDV-infected pigs are dysfunctional: they do not secrete IFN-γ and are not cytotoxic for NK-sensitive targets; at the same time, dendritic cells stop secreting IFN-α [82]. This is possibly related to the much greater shedding of FMDV from pigs in the early phases of infection, compared with ruminants [92]. Interestingly, although viremia is observed in cattle, levels of IFN-α are lower than those in pigs but do not abruptly decrease, and there is neither lymphopenia nor immunosuppression of NK or γδT cells [82], which points at a better control of FMDV in the early stages of the infection.

Finally, the profile of the innate immune response is probably correlated with the onset of the carrier state in FMD-convalescent ruminants; viral persistence in the upper respiratory tract is correlated in fact with decreased levels of mRNA for several immunoregulatory cytokines in the FMDV-infected tissues [93].

#### 2.2.2. IFN Sensitivity of FMDV

FMDV is extremely susceptible to Type I IFNs in vitro, and it can be even employed for a bioassay of swine IFN-β on porcine IBRS-2, an established pig kidney cell line [94]. FMDV transiently replicates in porcine nasal mucosal and tracheal mucosal epithelial cell cultures without cytopathic effects; production of IFN-β and IFN-inducible gene Mx1 mRNA is correlated with the disappearance of viral RNA and progeny virus [95].

Among Type I IFNs, rHis-PoIFN-ω7 recombinant protein conferred significant in vitro protection against FMDV, including two strains belonging to type O and A FMDV, respectively [96]. Additionally, PoIFN-δ8 exerted a significantly protective effect against FMDV in IBRS-2 cells [97], and the same results were obtained with PoIFN-alphaomega (αω) both pre- and post-infection [98]. There is also evidence of a high sensitivity of FMDV to IFN-γ, at least in bovine cells [99]. There is also evidence of a synergism between IFN-α and IFN-γ in the control of FMDV infection. A recombinant Adenovirus co-expressed porcine IFN-α and IFN-γ in tandem and induced interferon stimulated genes (ISGs) related with IFN-α and IFN-γ in porcine kidney (IBRS-2) cells; the antiviral effects were enhanced compared to that of Adenovirus expressing only a single protein [100]. Recombinant BoIFN-λ3 as glycosylated secreted protein also exerts antiviral activity against FMDV in bovine cell cultures [101].

#### 2.2.3. Type I IFN Responses

As mentioned in the above section, Type I IFNs are both induced and antagonized by the virus. In vitro, induction of Type I IFNs is a bottleneck for virus isolation and neutralization assays. Accordingly, the lack of an IFN response underlay the advent of hamster BHK-21 and porcine IBRS-2 for large-scale diagnostic applications [102]. Even in competent cell cultures, Type I IFN responses are limited and generally do not prevent induction of thecythopathic effect (CPE). On the contrary, the O1Lif mutant (probably endued with a defect of the Lpro gene [102]) can sustain, in the presence of guanidine, a high-titered production of IFN-β in bovine kidney cells [103]. Instead, induction of IFN-β is severely antagonized by non-mutant FMDV strains by means of the L pro proteinase [104].

A high level of constitutive expression of Type I IFNs has been demonstrated in pigs [105,106]. This has important implication in the innate immune response to FMDV. Thus, swine skin DCs express and store IFN-α in uninfected animals, which confers resistance to a subsequent FMDV infection [107].

Interestingly, high levels of type I IFN are produced in vitro after stimulation of pDC with FMDV immune complexes; the presence of an intact, infectious viral RNA is mandatory for IFN induction; additionally, in vivo, pDC are probably the major source of type I interferon production during acute FMDV infection [108]. FMDV immune complexes correlated with the favorable, long-lasting Ab response to an experimental FMD vaccine in calves [109], and with the greater protection observed in FMD-vaccinated guineapigs [110].

#### 2.2.4. Type II IFN Responses

Virus-induced Th1-like cytokine protein and mRNA (IFN-γ and IL-2) can be detected in pigs after injection of high-potency, emergency FMD vaccines [84]. This is in agreement with the in vitro IFN-γrecall response of whole blood samples from FMD-vaccinated cattle [111]. There is also a positive correlation in cattle between in vitro IFN-γ recall response and FMD vaccine-induced protection as well as reduced virus persistence; CD4+ T cells play a major role in this recall response, and the combination of the IFN-γ response with virus neutralizing antibody titers represents a better correlate of protection [112]. After FMDV infection, the peak expression of IFN-γ was observed in cattle two weeks after disease outbreak, the titers being much higher in clinically infected animals compared with the subclinical ones [113].

#### 2.2.5. Type III IFN Responses

There is evidence of a Type III IFN response in FMDV-infected cattle, with a possible important role in virus clearance; in fact, the level of expression of IFN-λ mRNA was higher in follicle-associated epithelium of the dorsal soft palate and dorsal nasopharynx of animals that had cleared the infection, compared with persistent virus carriers [114]. No such data are available in pigs, whichdo not become persistent FMDV carriers [78].

#### 2.2.6. Translational Prospects

There is evidence in vivo of IFN-mediated clinical and virological protection. For instance, Ad5-delivered type I IFN can rapidly protect swine against several FMDV serotypes [115]. Additionally, a constitutively active fusion protein of porcine IFN regulatory factors (IRF) 7 and 3, delivered with a replication-defective adenoviral vector, protected pigs from FMD clinical signs and viremia [116]. Pigs pretreated with 30 mg rPoIFN-γ were completely protected from virulent FMDV challenge, whereas lower doses delayed the appearance of clinical signs [117]. IFN-α also showed interesting properties as adjuvant of an experimental FMD vaccine containing the swine IgG single heavy chain constant region and an immunogenic peptide of serotype O FMDV. This resulted in strong induction of FMDV-specific neutralizing antibody and significant T-cell-mediated immune responses, as well as in the protection of vaccinated pigs [118]. Swine inoculated with a combination of Ad5-Pifn-α and Ad5-A24 subunit vaccine and challenged 5 days later were all completely protected from disease and developed a significant FMDV-specific neutralizing antibody response [119]. Adjuvant activity can also be exerted by IFN inducers. Pigs co-immunized with B4 (the G-H loops of three topotypes of FMDV serotype O and promiscuous artificial Th sites) and polyinosinic-cytidylic acid [poly(I:C)] were completely protected following FMDV challenge; B4 and poly(I:C) elicited FMDV-specific neutralizing antibodies, total IgG antibodies, type I interferon (IFN-α/β) and IFN-γ [120].As for Type III IFNs, Ad5-boIFN-λ3 delayed disease onset for at least 6 days when cattle were challenged 24 h after the treatment by intradermolingual inoculation of FMDV; the time period increased to 9 days when treated cattle were challenged by aerosolization of FMDV [121]. IFNs show some synergism with antiviral drugs such asAd-siRNA or ribavirin in IBRS-2 cells and suckling mice; therefore, the combination of Ad-porcine IFN-α and Ad-siRNA or ribavirin may represent a convenient strategy to overcome FMDV resistance against antiviral agents [122].

FMDV is highly susceptible to the effector mechanisms of innate immunity. This is the reason why the virus has evolved highly sophisticated strategies to circumvent such responses and, in particular, to block and/or reduce an effective IFN response. These outright decoy strategies are a foundation of FMDV virulence and rapid spread among susceptible animals. As for pigs, the early curtail of Type I IFN responses and NK cell activities account for a huge replication and dissemination of FMDV in the early stages of infection. A crucial role of the innate immune response can also be envisaged in the onset of the carrier state in FMD-convalescent ruminants.

While a plethora of studies has been devoted to the adaptive immune response to FMD vaccines, fewer investigations dealt with the corresponding innate immune response, which is likely to account for the early protection induced by potent FMD vaccines. Finally, the high IFN sensitivity of FMDV can underlie interesting translational prospects, in which pre-formed IFNs and/or viral vectors coding for IFN molecules could be pivotal to the early control of FMD outbreaks, and complement the use of suitable emergency FMD vaccines.

### 2.3. Porcine Coronaviruses

Coronaviruses (CoVs) are pleomorphic, enveloped and 60 to 220 nm in diameter, including the spike (S) glycoproteins that are approximately 12 to 25 nm in length. CoVs are single-stranded, positive-sense RNA viruses [123]. CoVs exist asquasispecies and have high rates of mutation and recombination [124,125]. This favors the emergence of new CoV strains with altered host specificity and cell tropisms, propelling their interspecies transmission to infect new animal and human hosts. The family Coronaviridae in the order Nidovirales is composed of four genera: Alphacoronavirus, Betacoronavirus, Gammacoronavirus and Deltacoronavirus [126].

Currently, six CoV can infect pigs, including four alpha coronaviruses, namely Transmissible Gastroenteritis Virus (TGEV), Porcine Respiratory Coronavirus (PRCV), Porcine Epidemic Diarrhea Virus (PEDV) and Swine Acute Diarrhea Syndrome-Coronavirus (SADS-CoV), as well as one betacoronavirus and one deltacoronavirus, namely porcine hemagglutinating encephalomyelitis virus (PHEV) and porcinedeltacoronavirus (PDCoV), respectively [126]. PRCV, a naturally occurring respiratory deletion mutant of TGEV with deletions in the S protein, was first isolated in Belgium in 1984 [127]. It causes mild respiratory disease, such as coughing, but no enteric disease such asthe parental TGEV. Since the emergence of PRCV, the spread of TGEV has also been reduced in PRCV-seropositive herds due to cross-protective immunity to TGEV [128].

#### 2.3.1. Immune Response to PRCV

The innate immune response is critical for host defense against respiratory CoVs [129]. Most CoVs are sensitive to the antiviral effects of virus-induced alpha/beta interferon (IFN-α/β). Specifically, the group 1 CoVs in the family Coronaviridae, order Nidovirales, PRCV and TGEV, are potent IFN-α inducers [130,131]. Specifically, the innate immune responses of the host against respiratory viruses involve alveolar macrophages, pulmonary epithelial cells, natural killer (NK) cells, dendritic cells and IFN-α/β responses in the lung. These responses influence the initial virus infection and also regulate adaptive immune responses. It has been reported that PRCV is a good IFN-α inducer [132,133,134].

Interestingly, it has been reported by Van Reeth and Nauwynck [135] that a subclinical course of the infection together with a gross lung consolidation involving up to 34% of the investigated lungs at 4 days post infection (DPI) was associated with high levels of IFN-α, while TNF-α was negligible and IL-1 even undetectable. Subsequently, Charley and colleagues [130] confirmed that type I IFN in pigs after experimental infection with PRCV started to be produced in the lung secretions within 24 h after infection, and it lasted for more than 4 days, in the absence of significant levels of pro-inflammatory cytokines such as TNF-α and IL-1. They reported also very mild clinical signs and a very mild lung neutrophil infiltration, which suggests that type I IFN in itself is not involved in pro-inflammatory and harmful effects [135]. Moreover, in line with the previous reported findings, in another study from Atanasova and colleagues [136], after PRCV experimental infection, high titers of IFN-α and IL-6 were found in the lung secretions of all the pigs under study between 1 and 5 DPI, after which the titers decreased, and by day 15, IFN-α and IL-6 were no longer detectable. Another study from Zhang and colleagues [131] also reported, between 4 and 10 days post PRCV infection, increased levels of IFN-α in bronchoalveolar lavage (BAL) and serum. Increased IFN-α was followed by increased IFN-γ, and IL-4 levels in BAL and in serum. In addition, high IFN-γ levels were found from 1 to 7 DPI, whereas only very low levels were detected between 9 and 15 DPI. Similarly, IL-12 levels in the BAL fluids peaked between 1 and 5 DPI and slowly decreased until 15 DPI when they were no longer detectable [131].

We can conclude that PRCV induces a different cytokine profile when compared to primary respiratory viral pathogens such asswine Influenza virus [43,137,138]. The latter, after intra-tracheal inoculation induces a higher and faster production of IFN-α, IL-1, IL-6 and TNF-α, and this may explain the much more prominent respiratory disease compared to PRCV [43,137].

#### 2.3.2. Translational Prospects

Interestingly, PRCV infection in pigs has been suggested as a reliable model for studying the effect of corticosteroids on Severe Acute Respiratory Syndrome Coronavirus (SARS-CoV) infection in humans [139]. It has been reported that the PRCV-induced increases in Th1 and pro-inflammatory cytokines have some similarities to cytokine profiles observed in SARS patients. In humans infected with SARS-CoV, elevation of IFN-γ and IL-6 in the blood was observed [140,141]. The similarities in cytokine profiles between PRCV infection in pigs and SARS patients further supports PRCV as a potential model for respiratory infections resembling SARS [131].

SARS-CoV causes severe illness in humans and is lethal in 10% of the cases, in contrast to PRCV infection in pigs, which is usually mild or subclinical; yet, there are some similarities between the courses of these two infections [136]. SARS-CoV starts as a mild pneumonia and clinical worsening typically appears at a late stage of infection (>1 week), when the virus load in the lungs starts to decrease and virus-specific antibodies appear in the circulation [136]. It has been demonstrated that respiratory dysfunction is caused by diffuse alveolar damage (DAD), which is believed to be triggered by an immunopathological reaction [142,143]. Similarly, in PRCV, pigs show pneumonia and DAD at 7–9 DPI, when antibodies are already detectable in the blood [136].

Importantly, Cameron and colleagues [144] suggested that dysregulation of innate and adaptive immune responses due to loss of homeostasis of type I (IFN-α) and type-II (IFN-γ) IFNs lead to a failure of SARS-CoV clearance from the lungs in severe SARS patients at crisis, whereas resolution of IFN and IFN-stimulated gene response in non-severe SARS patients was associated with recovery.

Interestingly, PRCV shows similar features to both SARS-CoV and SARS-CoV-2, such as tropism for both the upper and lower respiratory tract [128]. PRCV causes bronchoalveolitis as a result of necrosis of epithelial cells lining the upper and lower respiratory tract, followed by type 2 pneumocyte hyperplasia and hypertrophy and infiltration of macrophages and lymphocytes, causing thickened alveolar septa [139] and increased inflammatory responses, including IFN-α, TNF-α, IL-6, IFN-γ and IL-12 in the lung [136], similar to SARS-CoV-2 or SARS-CoV patients [145,146,147,148]. Accordingly, as the pulmonary pathology of PRCV infection in pigs resembles SARS in humans, PRCV infection of pigs was previously suggested as a model to examine SARS [139,149].

In comparison, there is much less systemic pro-inflammatory cytokine responses in PRCV-infected pigs, consistent with mild or subclinical disease [128]. In addition, unlike PRCV infection, neutrophils frequently infiltrate at the infection sites in the lung or in the blood of COVID-19 patients [150]. It has been stressed that the respiratory disease due to non-complicated PRCV infection in pigs is mild and self-limiting, similar to COVID-19 in most SARS-CoV-2 infected individuals [149].

Many interesting in vitro approaches have been described as a reliable alternative to animal models for investigating susceptibility to SARS-CoV-2; these include nasal mucosa explant (NME), air-liquid interface (ALI) cultures and precision cutting lung slides (PCLS), as recently reviewed by Runft and colleagues [151]. The pig is an interesting case, as two different porcine cell lines were found to be permissive to SARS-CoV-2 infection and showed cytopathic effects (CPE) [149,152]. Furthermore, its ACE2 protein was both predicted [153] and demonstrated experimentally [148,154] to bind to the SARS-CoV-2 spike protein. However SARS-CoV-2 neither replicated nor caused disease in pigs after experimental infection [155,156]. Meekins and colleagues confirmed that pigs seem resistant to SARS-CoV-2 infection, despite clear susceptibility of porcine cell lines. Pigs are, therefore, unlikely to play an important role in the COVID-19 pandemic as virus reservoir, nor are they likely to be a suitable pre-clinical animal model to study SARS-CoV-2 pathogenesis or to develop disease control measures.

Interestingly, Meekins and colleagues [152] reported that all pig studies to date have used rather young and healthy pigs, as well ascommercially available pig breeds/genetics. Co-morbidities, different breeds andincreased age could possibly make pigs more susceptible to infection. Additionally, unpredictable genetic changes in the SARS CoV-2 genome might result in a better compatibility of the virus for pigs in the future [152].

It is well accepted that alternative pre-clinical animal models, namely non-human primates [157], Syrian hamsters [157], transgenic or transduced mice expressing human ACE2 [157], ferrets [158] or even cats [159] need to be considered to gain additional insights into SARS-CoV-2 pathogenesis and virulence. Neutralizing antibody responses were detected in pigs infected via intramuscular or intravenous inoculation [160]; this indicates that pigs could be used for immunogenicity studies related to SARS-CoV-2.

Therefore, Graham and colleagues [161] developed a pig model for generating immune responses to vaccination against SARS-CoV-2. The inherent heterogeneity of an outbred large animal model is likely to be more representative of immune responses in humans. The data shown by Graham and colleagues [161] elegantly demonstrate the utility of the pig as a model for evaluation of the immunogenicity of ChAdOx1 nCoV-19 and other COVID-19 vaccines. In this study, T cell responses were higher in pigs that received a prime-boost vaccination regime, when compared to prime only at day 42, whilecomparing responses 14 days after last immunization demonstrated that the prime-boost regimen trended toward a higher response [161].

Finally, it is worth reporting another interesting model mentioned by Heegard and colleagues [149] based on obese Ossabaw pigs. Severe COVID-19 may be faithfully reproduced in PRCV-infected pigs co-affected by an underlying, chronic condition such an obesity-associated metabolic syndrome. The hypothesis underling this model is that disease severity will increase in obese Ossabaw pigs infected with PRCV compared to pigs of normal weight, and hence, will constitute a useful model for severe COVID-19 in humans at risk due to metabolic syndrome associated comorbidities, including aged individuals [149]. In order to provide a One Health perspective on disease mechanisms related to SARS-CoV-2 infection and the potential viral or host factors that contribute to the severity of COVID-19, it is critical and very important to understand the animal CoV infections in the natural host.

## 3. IFN Responses to DNA-Viruses

In this section, we deal with the peculiarities of the IFN response to three DNA-viruses, sustaining severe disease and heavy economic losses worldwide.

### 3.1. Porcine Circovirus 2

The Porcine Circovirus Associated Disease (PCVAD) complex refers to clinically severe disease conditions of swine [162]. PCVAD is caused by Porcine Circovirus 2 (PCV2), a small, non-enveloped DNA virus of the family Circoviridae [163]. PCV2 has been a ubiquitous, inoffensive viral agent over many years [164]; the advent of lean type, rapid growth pig phenotypes has been probably pivotal to transforming a subclinical infection into a devastating swine disease. This was indirectly confirmed by genomic analyses of PCV2, showing a high degree of homology between PCV2 strains from PCVAD-affected and healthy farms, respectively [165]. Postweaning multisystemic wasting syndrome (PMWS) is most common within the PCVAD complex. Affected pigs show wasting, respiratory signs, jaundice, interstitial pneumonia, lymphopenia and enlarged lymph nodes, in which lymphocytes (mainly the B cell areas) are largely depleted and histiocytes accumulate [166].Generalized lymphocyte depletion in primary and secondary lymphoid tissues is a prominent feature in clinically affected pigs. Other reported disease conditions are porcine dermatitis and nephropathy syndrome (PDNS), proliferative and necrotizing pneumonia (PNP), necrotizing tracheitis, congenital tremors, fetal myocarditis and reproductive failure [167]. PCV2 gives rise to PCVAD in swine co-infected by other viral agents, and/or exposed to environmental, non-infectious stressors (related e.g., to stocking density, diet andtemperature changes) [168]. A striking correlation has been observed between viral load in tissues and the severity of disease expression [169].

A wide variability in the viral genome led to the appearance of several new PCV2 strains worldwide. As a result, two original main groups were identified, i.e., genotype “a” (PCV2a) and “b” (PCV2b); further genotypes were subsequently reported in different areas of the world (PCV2c, PCV2d andPCV2e) [170,171]. Most commercial vaccines are based on PCV2a, which provides cross-protection against other PCV2 genotypes [172]. Accordingly, PCV2 genotypes are included in one single serotype in terms of vaccine coverage [173]. The current PCV2 vaccines are based on either inactivated whole viral particles, or ORF2 capsid (Cap) protein, assembled into virus-like particles (VLPs) [170]. Since PCVAD is hardly reproducible under experimental conditions, vaccine efficacy is usually tested in terms of reduction/inhibition of post challenge PCV2 viremia [174].

#### 3.1.1. Innate Immune Response to PCV2

Viral antigens are mainly found in the cytoplasm of diverse myeloid cells, such as histiocytes, multinucleated giant cells and other cells of the monocyte/macrophage lineage, even though epithelial cells probably play a major role for virus replication in vivo [167]. The tropism of PCV2 for myeloid cells underlies a potential to affect crucial circuits of innate immunity. PCV2 shows in fact a notable immunomodulatory capacity, deeply affecting the innate immune response of swine. PCV2 enters both conventional and plasmacytoid dendritic cells (cDC and pDC, respectively); whereas cDC function is not impaired, pDCs are deeply affected by infectious PCV2, whichcan down-regulate in vitro the induction of IFN-α in cultured pDC and monocytes [175]. After PCV2 infection, PMWS-affected animals are not able to mount effective, proinflammatory cytokine responses, as opposed instead to vaccinated animals in the early stages of the infection [176]. On the whole, there are opposite actions on pDC of (i) PCV2 encapsulated ssDNA (stimulatory) and (ii) free dsDNA replicatory forms (inhibitory) [177]; this is probably the reason why the stimulatory effects do prevail in case of whole virion-inactivated PCV2 vaccines. Accordingly, infectious PCV2 is a potent inducer of IL-10 in PBMC in vitro, which inhibits IFN-γ and IL-2 production by a recall antigen [178]. The use of inactivated PCV2 vaccines has led to reduced numbers of co-infections on farms [179]; these non-specific effects might be due to a mechanism of “Trained Immunity”, acting on innate immunity genes [180], and/or due to general immunostimulatory properties of inactivated PCV2 virions and VLPs.Thiswas reported, e.g., for Calicivirus VLPs, able to activate immature porcine bone marrow-derived dendritic cells in vitro [181]. Vice versa, similar, non-specific protective effects against PCVAD were observed in parvo/erysipelas-vaccinated sows on farms [182].

#### 3.1.2. IFN Sensitivity of PCV2

Contradictory findings were obtained in studies about IFN sensitivity. On the one hand, IFN-γ added before, during or after inoculation significantly increases the number of PCV2-positive PK-15 and porcine monocytic cells (3D4/31 line); the enhancement of PCV2 infection was induced by IFN-α only after PCV2 inoculation; effects were shown to be quite specific in that they were blocked by IFN-specific, neutralizing antibodies; a 20-times higher PCV2 production was obtained in IFN γ-treated PK-15 cells as a result of increased internalization of viral particles [183]. On the contrary, IFN-α added before PCV2 inoculation decreased the number of virus-infected PK-15 cells [183]. PCV2 infection also induces a strong interferon IFN-β response in the VR1BL porcine fetal retina cell line, which also facilitates viral gene expression; this may be due to the presence of an interferon-sensitive response element in the viral promoter [184], as described in a previous study [185].

#### 3.1.3. Type I IFN Responses

As statedabove, PCV2 can enter pDCs and affect their ability to release type I IFN through TLR-7 and TLR-9 receptors following infection with Classical Swine Fever virus (CSFV), Transmissible Gastroenteritis Virus (TGEV) and Pseudorabies Virus (PRV), as recently reviewed [186,187]. This inhibition of the IFN-α response was linked to the DNA of the virus, not to PCV2 replication within pDC [188]. Despite the inhibitory control actions on pDC, an IFN-α response to PCV2 infection has been repeatedly detected in vivo; after the IFN-α response, elevated levels of serum C-Reactive Protein and IL-10 in the secondweek after infection were associated with PCV2-infected piglets that subsequently developed severe PMWS [189]; additionally, a clear correlation was observed between viral load and IL-10 amounts in vivo [190]. This time-course of IL-10 is clearly reminiscent of the clinically severe PRRS cases referred to in a previous chapter [52,53,54]. Induction of both IFN-α and IL-10 was also observed in pigs infected with porcine parvovirus in combination with PCV2 during experimental reproduction of PMWS [191]. An IFN-α response was detected in vitro in cultures of swine alveolar macrophages, and the released IFN-α was shown to inhibit the cytopathic effect of PRRSV on MARC-145 cells and alveolar macrophages [192]. PCV2 induces IFN-β in PK-15 cells via the RIG-1 and MDA-5 signaling pathways, which is positively related to increased viral replication [193].

#### 3.1.4. Type II IFN Responses

The PCV2-specific IFN-γ response in whole blood samples is a robust correlate of protection after vaccination with inactivated, whole virion vaccines [174]. Reduction of PCV2 viremia is associated to both PCV2-specific neutralizing antibodies (NA) and interferon-γ-secreting cells (IFN-γ-SCs) in PCV2-vaccinated animals; vaccination with an inactivated, chimeric PCV1-2 vaccine also gives rise to a delayed type hypersensitivity (DTH) response only in vaccinated animals [194]. A DTH response was also observed in pigs injected with an inactivated, whole virion PCV2 vaccine, as opposed to unvaccinated, PCV2-infected animals [194]. PCV2-specific, IFN-γsecreting cells were also observed following injection of a VLPs-based vaccine [195]. On the contrary, an experimental PCV2 vaccine containing free, non-assembled Cap protein of PCV2b failed to induce a PCV2 virion-specific IFN-γ response in whole blood samples and in a flow cytometry-based assay [196]. A central role in clearing PCV2 infection could be actually played by PCV2-specific, IFN-γ/TNF-α co-secreting CD4+ cells after vaccination and infection [195]. Our experiments indicated that the use of mock virus (cryolysate of PK15c28 cells) as negative control in the IFN-γrelease and ELISPOT assays was mandatory to obtain reliable results, because of significant overlapping of virus-specific and non-specific immune signals [174]. This phenomenon could be due to the recognition of stress antigens by NK and T cells [197]. In particular, stress antigens are expressed on PK15 cells and other established pig cell lines of epithelial and endothelial origin [198], as also confirmed in the authors’ laboratory [174]. Therefore, unless a highly purified Ag is employed to test the IFN-γrecall response to PCV2, a proper control Ag should be employed to discriminate between PCV2-specific and non-specific responses. Finally, a negative result in the IFN-γ release assay on whole blood does not rule out the presence of PCV2-specific effector T cells elsewhere. In fact, in the ‘‘hidden memories’’ scenario, long-lived effector memory T cells are present at sites of pathogen invasion and/or mobilized within hours to such sites from blood and marginated pools [199].

#### 3.1.5. Type III IFN Responses

Little information is available about type III IFNs in pigs. Two porcine type III IFNs, *Sus scrofa*IFN-λ1 (SsIFN-λ1) and SsIFN-λ3, have been described; these show antiviral activity and are probably implied in the modulation of the host–pathogen interaction [200]. To our knowledge, no specific data are available about type III IFN responses induced by PCV2 in vivo and in vitro.

#### 3.1.6. Translational Prospects

Because of the role of PRRSV co-infection in the onset of PCVAD, the treatment with an oral low dose of IFN-alpha resulted to be beneficial for farm [73]. Additionally, a beneficial role of these treatments can be reasonably envisaged because of an improved control of the serious, early weaning stress in IFN-α-treated piglets [201]. The aforementioned enhancement of viral replication by both type I and II IFNs substantially deterred from investigating these cytokines for either immunomodulation or improved adjuvant features of current PCV2 vaccines [183]. On the contrary, vaccine design has been oriented to an improved presentation of PCV2 Cap protein, reducing the in vivo IFN-γ response; accordingly, in the mouse model, a vaccine based on secreted Cap was associated to a higher neutralizing Ab response and reduced production of IFN-γ [202].

The onset of PCVAD does not seem related to peculiar pathogenicity markers of PCV2, but to a distinct susceptibility profile of currently farmed pigs. These often fail to effectively counter PCV2-driven effects on the homeostasis of the innate immune response, which paves the way in turn to the onset of severe PCVAD. In this respect, the inhibition of IFN-α responses in pDCs and the IL-10 response in PCV2-infected pigs are likely to play a crucial role in making the loss of homeostasis irreversible, as a foundation of different forms of PCVAD.

Despite a clear inhibitory control of pDC functions, Type I IFN responses to PCV2 infection are induced in other cell types, with a possible major role of some macrophage populations. These are susceptible to PCV2 entry; however, they play a minor role for virus propagation.

A protective role of PCV2-specific, IFN-γ SC has been repeatedly highlighted. This is definitely at odds with the observed enhancement of virus replication in the presence of IFN-γ, as well as with the clear correlation between viral load in tissues and onset of PCVAD. How can one reconcile such diverging findings? First, it should be stressed that the IFN-γ response has been validated in experimental trials as a correlate of protection against PCV2 viremia, but not against PCVAD. Therefore, a discrepancy might be envisaged between clinical and virological protection, in which the IFN-γ response might play quite different roles. Secondly, the overall balance of cytokine responses might be quite different if protection were to be related to PCV2-specific, IFN-γ/TNF-α co-secreting CD4+ cells [195]. Thirdly, last but not least, IFN-γ could check the onset of the IL-10 response, as shown in the PRRS model [69]; this could prevent an irreversible loss of immune homeostasis, underlying different forms of PCVAD. Please notice that IL-10 may show a pro-inflammatory gain within an established inflammatory environment, as previously shown in human models of endotoxemia [203] and Crohn’s disease [204]. Additionally, IFN-α and IL-10 can also co-promote the differentiation of Type I T regulatory (Treg) cells [58].

On the whole, little if any direct antiviral role is probably played by IFNs in the control of PCV2 infection. On the contrary, both Type I and Type II IFNs are probably a double-edged sword: they are involved in the pathogenesis of PCVAD, as well as in induction and differentiation of antiviral immune effector activities.

### 3.2. Aujeszky’s Disease

Aujeszky’s Disease (AD) or Pseudorabies is a viral disease of several animal species caused by Suidalphaherpesvirus 1, also known as Pseudorabies virus (PRV). The disease was described first by AladàrAujeszky in 1902. Only pigs are able to survive an established PRV infection, and they are considered as the natural host of the virus. Pigs exhibit a pronounced age-related resistance, with younger animals being more susceptible to fatal infections characterized by neuronal signs, followed by sudden death. In contrast, older animals primarily present with respiratory distress or even subclinical infection. In pregnant animals, infection of fetuses results in resorption, mummification or abortion [205]. AD became of concern in the 20th century, as pig farming activities and their intensification grew to unprecedented levels. A major breakthrough in disease control was the advent of live-attenuated vaccines in the early 1960s. Later on, in the 1980s, large deletions in the genome of such PRV vaccine strains were detected, which allowed for the development of marker vaccines and related DIVA procedures [205]. Nowadays, these underlie disease control and eradication efforts at the regional level in many areas of the world.

#### 3.2.1. Innate Immune Response to PRV

Innate immunity plays a major role in regulating the sensitivity of animals to PRV infection. In particular, Type I IFNs are of paramount importance, since α/β interferon receptor-deficient mice are much more susceptible to virulent Pseudorabies virus infection [206]. With an attenuated gE-TK-PRV strain, mice with an intact γ IFN system (but without mature B and T cells) were able to prevent systemic virus dissemination [207]. On the whole, there is strong evidence that different pathways of innate immune response are triggered after PRV infection; these are submitted to epigenetic regulation with a major role of histone deacetylases (HDACs). Inhibition of HDAC1 significantly checks PRV replication by inducing a DNA damage response and antiviral innate immunity through the downstream STING/TBK1/IRF3 innate immune signaling pathway [208]. The same epigenetic effect can be obtained by using inhibitors of bromodomain-protein 4 (BRD4) [209]. PRV performs diverse inhibitory control actions on the innate immune response, which underlies effective virus replication and virulence. In primary rat fibroblasts, wild-type PRV checks the expression of genes normally induced in IFN-β-treated cells, because of a less effective phosphorylation of STAT1 [210]. In addition to that, PRV UL13 directly inhibits cGAS-STING-mediated IFN-β production and the expression of multiple interferon-stimulated genes (ISGs) following IRF3 phosphorylation [211]. Additionally, other structural and non-structural proteins of PRV can down-regulate crucial pathways of the innate immune response, including UL24 [212], the UL13 protein kinase [213], dUTPase UL50 [214] and UL42 [215]. All of these inhibitory control actions aim to check Type I IFN responses, which highlights their importance for the control of PRV infection.

#### 3.2.2. IFN Sensitivity of PRV

The sensitivity of PRV to IFNs has been described for a long time [216]. We also had observed in vitro sensitivity of the attenuated PRV BUK strain to porcine IFN-βby means of an in vitro PRV yield reduction assay [94]. In porcine nasal mucosal explants, recombinant porcine methionyl-IFN-α1 limited PRV infection to epithelial cells, with a reduced virus replication; at the same time, it protected the underlying stromal fibroblasts [217]. The various results obtained in different laboratories can be partly accounted for by the widely different antiviral activities within porcine Type I IFNs. Thus, the different subtypes of porcine IFN-α showed dramatic differences in antiviral activity against PRVBartha BS/K61 in bovine MDBK cells treated with 10 IU/mL of each porcine IFN-α subtype; IFN-a2, -a5, -a9, and -a10 showed the highest level of activity in a PRV yield reduction assay. On the contrary, little if any activity was shown by IFN-a3, -a7, -a13, -a4 and -a15 [4]. Accordingly, minor differences of the aminoacid sequence can account for dramatic increases in antiviral activity: the mutant PoIFN-α-156s inhibits much more PRV replication in a dose-dependent manner in vitro, compared with the original PoIFN-α [218]. Among IFN-stimulated genes, ISG15 [219] and ISG20 [220] probably play a major role in regulating the antiviral control activities. On comparing porcine IFNs, porcine IFN-β showed the highest activity in vitro; porcine IFN-γ was less effective and porcine IFN-α had the least capability of the three PoIFNs, whereas a combination of IFN-γ and IFN-α or β showed a strong synergistic inhibition of Pseudorabies virus replication [221]. Among Type I IFNs, in vitro, IFN δalso exerts a significant activity against PRV [222]. IFN-α may also promote efficient PRV latency establishment, as shown in an in vitro model of cultivated pig trigeminal ganglion neurons [223]. In this respect, the IFN-α response may be a double-edged sword: it can effectively counteract an acute PRV infection, but also be conducive to PRV survival in the host for a possible, stress-related reactivation later on.

#### 3.2.3. IFN Responses

Neither virulent, nor avirulent PRV strains could induce IFN in pig kidney cells, as opposed to chick embryo fibroblast cultures; in this case, IFN production was positively affected by aging of cells and negatively by virulence of the PRV strains under study [216]. On the contrary, a substantial IFN-α response can be elicited by PRV in vitro from purified porcine plasmacytoid dendritic cells, along with varying amounts of IFN-γ [224]. Once again, attenuation of PRV (see strain Bartha K/61) can be associated to an increased IFN-α response, which is accounted for by deletions in the gE/gI glycoprotein complex [225]. In particular, the inhibitory control of gE on IFN-β production takes place via CREB-binding protein (CBP) degradation [226]. Indeed, the IRF3–CBP/p300 interaction is crucial for IFN transcription so the CREB-binding protein degradation is associated to IFN-βdown-regulation. Infectious PRV is not mandatory for interferon induction. Intradermal administration of glutaraldehyde-fixed PRV-infected autologous or allogeneic cells gives rise to an interferon (IFN)-α response in pigs, detected in blood after a few hours; additionally, IFN-α/β mRNA containing cells can be demonstrated in regional lymph nodes [227]. In vitro, using the same IFN inducer, a genetic influence and stress (transportation and mixing) were shown to affect the ability to produce IFN-α in whole blood cultures [228]. PRV infection leads to an early differentiation of PRV-specific, IFN-γ secreting cells. In this respect, IFN-γ production by primed porcine PBMC can be a reliable and specific ex vivo marker of PRV infection, even before antibody formation [229]. A recall IFN-γ response in vitro was also observed in our laboratory in whole blood cultures of PRV-vaccinated pigs [230].

#### 3.2.4. Translational Prospects

On the whole, there is ample evidence of an effective control of PRV replication in vitro by different porcine IFN molecules. Yet, indications of effective therapeutic and prophylactic treatments in vivo are still limited and restricted to laboratory rodents. Thus, intraperitoneally and intracerebroventricularly given IFN-γ completely protected rats against a lethal PRV infection [231]. As in other models, IFN-γ shows a potent adjuvant activity of PRV vaccines. A significantly increased neutralizing Ab response was observed in pigs that had received an inactivated PRV vaccine containing 10^4^ U IFN-γ; these animals were also better protected against virus challenge in terms of post challenge fever and average daily weight gain [232].

### 3.3. African Swine Fever Virus

African swine fever (ASF) is a highly contagious and hemorrhagic disease affecting domestic pig and Eurasian wild boar. It is characterized by high mortality (up to 100%) and rapid death [233]. To date, it represents a worldwide threat to swine industry in many European and Asian countries, since there is no currently available vaccine [234]. It is actually present in several sub-Saharan African countries, Europe, Russian Federation, Trans-Caucasus, China, South-Est Asia, Oceania and, most recently, it was introduced in the Americas [235,236].

The causative agent of ASF is the African swine fever virus, which is the only member of the Asfarviridae family; it is a large, enveloped, linear, non-segmented and double-stranded DNA virus with ichosahedral morphology, 175 to 215 nm in diameter [237]. Depending on the virus strain, its genome varies between 170 and 190 kbp and contains between 150 and 167 open reading frames (ORFs) encoding for proteins essential to viral replication, assembly and host immune response evasion [233,238]. The main targets of the infection are monocytes and macrophages, which play a pivotal role in viral persistence and dissemination [239]. Currently, 24 genotypes have been identified by sequencing the major capsid protein (p72) gene (B646L); in particular, genotype I and genotype II have been reported outside Africa, with the latter being the cause of the ongoing pandemic [233,240]. Nevertheless, genotype I was recently described in China [241].

ASFV can infect domestic pigs, warthogs and bushpigs in Africa and wild boar in Eurasia; in Africa, the infection is maintained through a sylvatic cycle involving African wild suid species, soft ticks of the genus *Ornithodoros*(*O. moubata*) and domestic pigs, whereas in Europe, the cycle involves wild boar and domestic pig, with transmission occurring mainly through direct contact or ingestion of contaminated food [242].

Depending on the viral strain, the age and the immune system of the animal, clinical signs can vary: highly virulent strains lead to a peracute form (up to 100% mortality) characterized by high fever, cutaneous hyperemia and peracute death in 1–4 days P.I.; no evident lesions within the organs are present. Moderately virulent strains cause hemorrhagic fever, abortion in sows and other non-specific symptoms such as gastrointestinal and respiratory signs (up to 30–70% rate of mortality); common anatomopathological findings include hyperemic splenomegaly, cyanosis, enlarged hemorrhagic lymph nodes andpetechiae in the kidneys, intestine and gall bladder [234,243]. Low virulent strains can lead to subclinical and chronic disease with a lower rate of mortality [234].

The development of an effective vaccine for ASF still remains a challenge, and it is undoubtedly necessary to achieve a better understanding of the host’s immune system in order to make a step forward.

#### 3.3.1. Innate Immune Responses to ASF

As previously described, innate immunity represents the first line of defense against pathogens [2,3,4]; likewise, ASFV has evolved multiple strategies in order to evade and modulate the host’s innate immune system. In fact, the only possible way to design an effective vaccine, in the first place, is through a better characterization of the mechanisms underlying the host–pathogen interaction and orchestrating immune system evasion [244,245,246].

The main targets of ASF are mononuclear phagocytes of the myeloid lineage, including monocytes, macrophages and dendritic cells (DCs) [245,247]. Several studies demonstrated that the interaction of these cells with ASFV and their susceptibility to the infection can vary depending on the virulence of the isolate [239,245].

Garcia Belmonte and colleagues highlighted the role of the cGAS-STING-IRF3 route in ASFV infection and the differences in innate immune response between virulent and attenuated ASFV strains, by demonstrating that the attenuated NH/P68 strain activates the cGAS-STING-IRF3 cascade early during infection, resulting in the induction of significant levels of IFN-β in infected macrophages, whereas virulent Arm/07 strain inhibits IFN-β synthesis by modulating STING phosphorylation and subsequent IRF 3 activation [248]. These results are in agreement with previous works [249,250]. Macrophages are thought to be pivotal in viral persistence and dissemination: several in vitro studies focused their attention on the interaction between ASFV and these cells; differentiated macrophages (Mφ) proved to be more susceptible to ASFV infection in comparison to monocytes [251]. Franzoni and colleagues demonstrated that both virulent and attenuated strains were able to down-regulate the expression of CD14 (lipopolysaccharide, LPS, receptor) and CD16 (a low-affinity receptor for IgG Fc involved in antibody opsonization) in infected Mφ, leading to antiviral activity impairment; on top of that, they described that Mφinfected with attenuated NH/P68 and virulent 22653/14 strains released lower amounts of IL6, IL12 andTNFα in response to activation with IFN-γ and LPS, or a TLR2 agonist [252]. The same authors reported that, depending on the virulence and genotype of the strain, MHC class I expression can differently be modulated in infected Mφ: attenuated strains (BA71V, NH/68) and isolates belonging to genotype II (Arm07) or IX (Ken06.Bus) seemed to down-regulate MHC I expression on infected Mφ compared to virulent genotype I ASFV [245,246]; studies reported that, in vivo, this down-regulation could result in NK cell activation [253], thelatter being correlated to protection [254]. On the other hand, infection with both virulent and avirulent strains does not seem to affect MHC II expression in infected macrophages [252,255].

According to the literature, ASFV can lead to different chemokine/cytokine responses of Mφ after infection in vitro, based on the virulence of the isolate; virulent Georgia 2007 strain was shown to down-regulate the expression of anti-inflammatory cytokine (IL-10) and to up-regulate the expression of pro inflammatory cytokines of the TNF family (FASL, TNF, TNFSF4, TNFS10, TNFS13B andTNSF18) [256]. On the contrary, attenuated strains seemed to up-regulate the expression of key cytokines (IFN-α, IFN-β, ILβ1, IL12, IL18 and TNF-α) and chemokines (CCL4, CXCL8 andCXCL10), which could play a crucial role in the enhancement of immune responses [239]. Further studies are being conducted in order to better understand the response in polarized M1 and M2 Mφ by Franzoni and colleagues [247,252].

As for the dendritic cell (DC) response to ASFV, few studies have been conducted until now; in mid 1990s, Gregg and colleagues reported that ASFV could infect dermal DCs in vitro, subsequently impairing DCs infection by FMD virus; furthermore, in vivo, virulent ASFV L60 was able to infect and reduce the number of interdigitating DCs (iDCs) in mandibular lymph nodes at 3 days P.I., limiting the development of an effective and protective immune response [257,258]. More recent studies reported that in vitro infection of enriched blood DCs with ASFV led to the release of high levels of type I IFN; thus, it has been suggested that pDCs could represent the potential source of high levels of type I IFNs during acute ASF infection [259].

A recent study conducted by Franzoni et al. highlighted the differences in immune response between virulent (22653/14 ASFV) and attenuated (NH/P68) strains after in vitro ASFV infection of monocyte-derived DCs (moDCs). Both strains efficiently infected and replicated in moDCs, without inducing a strong cytokine response, but NH/P68 strain down-regulated MHC I, related to NK activation in vivo, similarly to what has been previously described in Mφ [245]. Further studies are needed in order to discern ASFV-DCs interaction during infection.

#### 3.3.2. Sensitivity to IFN

Contrasting data are present in literature as for ASFV sensitivity to IFN, since strain susceptibility could vary based on their virulence. Previous studies observed that both attenuated and virulent ASFV strains were sensitive to Bovine interferon-α 1 and porcine IFN-γ in both porcine monocytes and porcine alveolar macrophages (PAMs), with the most effective antiviral activity being exerted in IFN-γ treated PAMs [260]; a similar result was obtained in Vero cells pretreated with human IFN-α and then infected with attenuated BA71V [261]. A more recent study, by Golding and colleagues, observed that replication of virulent strains (BA71, Georgia 2007/1, OUR T88/1) was not inhibited in PAMs and blood derived macrophages treated with IFN-α, whereas the naturally attenuated strain NH/P68 and OUR T88/3 seemed to be more susceptible to high doses of Type I IFN; the enhanced sensitivity to IFN-α was associated to the lack of genes within the MGF360 and 505 regions [259]. Interestingly, Franzoni et al. [252] observedthat in vitro sensitivity to IFN-αwas dose-dependent; high doses of IFN-α (800 U/mL) led to a reducedpercentage of infectedmacrophages (Mφ) alsoafterinfection with virulent ASFV, independently of the genotype. However, Golding [259] and Franzoni [252] used different macrophage populations (PAMs and Mφ, respectively); this could explain the difference in the response to IFN-α. In another study regarding DCs, Franzoni et al. reported that, after maturation with IFN-α, treated moDCswereless susceptible to the infection with attenuated strains; in contrast to that, maturation with TNF-α resulted in an increased susceptibility to infection with virulent strains [245].

Several studies raised attention to the possible cause underlying a higher susceptibility of attenuated strains to IFN I. As a result, the induction of an antiviral state could be linked to ASFV MGF360 and 505 genome regions; indeed, it emerged that the deletion of genes within MGF360 and 505 regions from virulent genotype II Georgia 2007 or genotype I Benin 91/7 was associated to virus attenuation in vivo and induction of protective immunity [250,262].

#### 3.3.3. IFN Response

As previously discussed, the IFN system plays a crucial role in orchestrating innate immune response, by inducing an antiviral state in both infected and in neighboring, non-infected cells [239,244]. Indeed, the pivotal role of type I IFNs in controlling ASFV infection and inducing a protective immune response has already been demonstrated in vitro by several studies [260,261]. In light of this, it is not surprising that virulent ASFV strains have developed strategies to counteract IFN responses.

Studies conducted in vitro showed that, among ISGs, two interferon induced transmembrane proteins (IFTM2 and IFITM3) reduced BA71V virulent strain infectivity in Vero cells, by impairing endocytosis-mediated viral entry and uncoating [263], whereas MxA protein inhibited BA71V in Vero cells [264].

Several studies demonstrated that attenuated ASFV strains enhance type I IFNs expression, compared to virulent strains [249,265]. Razzuoli and colleagues, by conducting in vitro studies in porcine monocyte-derived macrophages (moMφ)unactivated and activated with IFN-α, both infected with NH/P68 and 22653/14 strains, highlighted a different type I IFN expression pattern, depending on the diverse virulence of the two strains; both strains induced the expression of some IFN-α subtypes (-α3 and -α7/11) in unactivatedmoMφ; however, compared to virulent 22653/14 strain, the attenuated NH/P68 strain induced a statistically significant up-regulation of IFN-α10, IFN-α12, IFN-α13, IFN-α15, IFN-α17 and IFN-β; interestingly, it has been reported by other authors that IFN-α3 and IFN-α7/11 showed no antiviral activity against PRV [4], PRRSV and VSV [9]. As for IFN-α-activated moMφ, both strains induced IFN-β and IFN-α subtypes expression, with NH/P68 strain significantly up-regulating IFN-α1, IFN-α10, IFN-α15, IFN-α16 andIFN-α17 [265]. Another recent study demonstrated that the attenuated NH/P68 strain activated the cGAS-STING-IRF3 signaling cascade early during infection, thus enhancing IFN-β expression in infected Mφ, compared to the virulent Armenia/07 strain [248].

According to the literature, ASFV virulent strains seem to have developed mechanisms to suppress type I IFN induction; differently, attenuated strains might have partially lost some of these immune evasion mechanisms [239,266,267,268,269]. Correia et al. demonstrated that ASFV-encoded multigene families (MGFs) inhibit the type I and II IFN response by targeting different intracellular signaling intermediates. The ASFV A276R gene (MGF360) inhibited the induction of IFN-β, by targeting IRF3, whereas ASFV A528R (MGF505-7R) was shown to target both IRF3 and NF-κB, thus limiting the type I IFN responses [270].Another viral gene implicated in the modulation of type I IFN induction is the one encoding the protein I329L, a TLR 3 homologue, able to inhibit the induction of IFN-β at the level of TRIF, by impairing the activation of both IRF3 and NF-κB [270,271].

Most recently, Reis and colleagues reported that deletion and interruption of several genes within MGF360 and MGF530/505 regions in virulent Benin 97/1 strain led to the induction of high levels of IFN-β mRNA in vitro, compared to the naturally virulent Benin 97/1 strain [250]. This wasconfirmed by a previously mentioned study [239,266,267,268,269], according to which virulent strains evolved multiple strategies, in order to evade and modulate the innate immune system, by limiting or impairing type I and II IFN responses.

Although many studies demonstrated that ASFV virulent isolates inhibit type I IFN responses in vitro, recent studies conducted in vivo reported divergent results; Karalyan and colleagues reproduced the infection in vivo withan ASFV virulent strain and measured the level of IFN-α, IFN-β and IFN-γ by ELISA assays: it emerged that while IFN-α reached the highest level at 2 days P.I. and then sharply decreased, the level of IFN-β and IFN-γ increased from 2 to 4 days P.I. [272].Additionally, Wang et al. reported a strong elevation of type I IFN at 3 DPI in pigs infected with type II virulent ASFV SY18 [273].Another study confirmed the presence of IFN-α and IFN-β in the serum of animals infected with virulent ASFV and also demonstrated the presence of a large amount of biologically active IFN several days before viremia reached its peak; thissuggested that IFN was not able to inhibit ASFV replication in vivo during viremia [259].

These conflicting results reported in the literature could be due to the differences between the types of macrophages effectively targeted by ASFV in vivo during infection and the porcine macrophages cell cultures available in vitro. In addition, a crucial role in type I IFN release in vivo might be played by pDC, as suggested by Golding [259]. Undoubtedly, further studies need to be carried out in order to better characterize the host–pathogen interaction and the mechanisms through which virulent and attenuated strains might modulate the innate immunesystem.

#### 3.3.4. Translational Prospects

To date, it is not clear whether IFN treatments could be used to prevent ASF infection: as previously described, in vitro studies demonstrated that human IFN-α, human IFN-γ and bovine IFN-α inhibit ASFV replication in Vero cells and Porcine Alveolar Macrophages PAMs, respectively [260,261]. However, in vivo studies regarding potential IFN treatments in infected pigs are still lacking.

Interestingly, a study recently conducted by Fan and colleagues demonstrated that 10^5^ U/kg of combined recombinant porcine IFNs (PoIFN-α and PoIFN-γ), administered to ASFV-challenged pigs, significantly up-regulated ISGs expression, reduced viral load, inhibited ASFV replication and alleviated clinical signs during the early stages of infection; thus, the authors suggested a potential antiviral role for recombinant porcine IFNs, indeed providing a new insight into ASF control strategies [274].

As for vaccines, the major difficulty in developing an effective one is mainly due to the complex nature of virus and the diversity in virulence between strains; until now, gene-deleted live attenuated vaccines (LAVs) proved to be efficient in controlling viral disease; because of the involvement of MGF genes in the inhibition of IFN production during ASFV infection, several authors have suggested that MGF360 and 505-deletedmutants could be used for immunization of pigs [250,262,270]. In particular, Reis and colleagues showed that pigs immunized and boosted with a MGF-deleted mutant of Benin 97/1 virulent strain were protected against challenge with a lethal dose of virulent Benin 97/1 [250].

At present, ASF remains a globally significant challenge to the swine industry; unfortunately, still little is known about the virus–host interaction, and a better understanding of the immune response generated during infection is needed, to provide new insights into ASF prevention and treatment strategies.

Although it has been established that the type I IFN system represents a crucial element in the innate immune response to ASFV infection, there are still many unknowns. Further in vivo and in vitro studies must be carried out in order to explore the several pathways through which ASFV strains can modulate the immune response and, particularly, IFN induction.

## 4. Conclusions

The huge variety of viruses affecting swine represents a global problem; indeed, every year, viral swine diseases cause serious economic losses in the swine industry. The IFN responses play a pivotal role in the immune response to viral agents (Table 1). Yet, we have also highlighted the constraints of the IFN response, mainly related to virus-induced immune escape mechanisms and the abnormally regulated “bad IFN-α response”, playing an important role in the PRRS and Influenza models. Therefore, it is essential to achieve a better understanding of the interaction between viral agents and the IFN system. This will lead the scientific community to major insights into the pathogenetic mechanisms of viral infections, toward improved therapeutic and preventive disease control measures.

## Author Contributions

Conceptualization, E.R., F.A., L.D.P., M.C. and M.A.; methodology, E.R., F.A., L.D.P., M.C. and M.A.; data curation, E.R., F.A., L.D.P., M.C. and M.A.; writing—original draft preparation, E.R., F.A., L.D.P., M.C. and M.A.; writing—review and editing, E.R., F.A., L.D.P., M.C. and M.A.; visualization, E.R. All authors have read and agreed to the published version of the manuscript.

## Figures and Tables

**Figure 1 pathogens-11-00175-f001:**
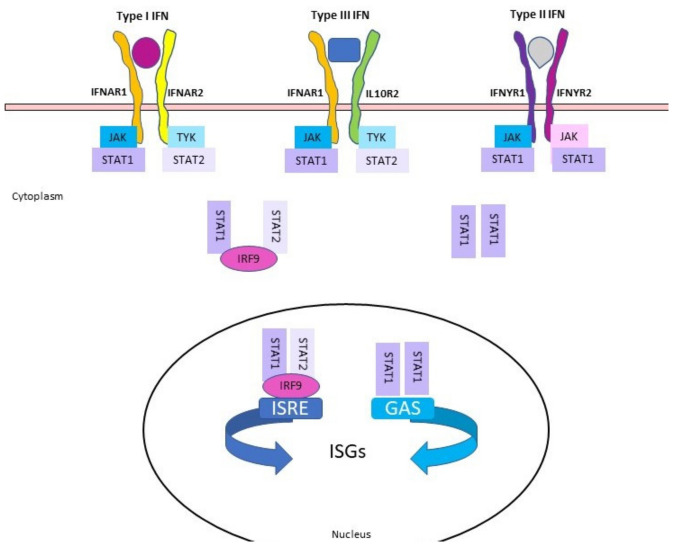
Simplified schematic of IFNs signaling cascades. Each IFN type has a cell surface receptor. Type I IFNs bind to IFNAR (composed by IFNAR1 and IFNAR2); type III IFNs bind to the IFNLR (consisting of IFNAR1 and IL-10R2) and type II bind to IFNYR (composed of IFNYR1 and IFNYR2). Binding of IFNs to their receptors cause cross-phosphorylation of JAK and TYK on the cytoplasmic domains of the receptor. This leads to phosphorylation of STAT1 and STAT2.

**Table 1 pathogens-11-00175-t001:** Major effects of viral infections on IFNs system.

	Type I and Type III IFN		Type II IFN	
*PRRSV*	In vivo early systemic induction of IFN-α response after challenge with PRRSV strains	[30]	In vitro detection of SC-IFN-γ secreting cells at 2–3 weeks P.I. showing an erratic behaviour	[61]
	In vivodown-regulation of IFN-α production following respiratory infection	[37]	In vivo transient and limited IFN-γ response after PRRSV infection	[66]
	Up-regulation of type I IFN gene expression by miR-26a	[44]	Early IFN-γ production in PRRS-infected pigs from NK cells activation	[68]
	Down-regulation of type I IFN gene expression by miR-373 and miR-382-5p	[45,46]		
	In vivo induction of a “bad IFN-α response” in PRRSV-infected pigs	[52,53]		
	Down-regulation of key transcription factor (IRF-1, IRF-3 andNF-κB) after ADE-mediated PRRSV infection	[60]		
	Reduced/unchanged IFN type III expression in PAM-pCD163 cells following nsp2-deletion mutants infection	[72]		
*FMD*	In vitro limited induction of a type I IFN response	[102]	In vitroinduction of IFN-γ response in whole blood saples from FMD-vaccinated cattle	[111]
	Up-regulation of IFN-β expression in bovine kidney cells following O1Lif mutant infection	[102,103]	In vivoproductionof IFN-γ after injection of high potency, emergency FMD vaccines in swine	[84]
	Down-regulation of IFN-β production after infection with non mutant FMDV strains	[104]		
	In vivo constitutive expression of Type I IFNs response	[105,106]		
	In vitroup-regulation of type I IFN production following stimulation of bovine pDC with FMDV immune complexes	[108]		
*PRCV*	Detection of high levels of IFN-α during subclinical course of PRCV infection	[135]	Detection of high IFN-γ levels IN BALs fluid and serum of PRCV infected pigs	[131]
	Type I IFN production in lung secretions within 24 h PI in experimentally infected pigs	[130]		
	Detection of high IFN-α levels in BALs fluids and serum of PRCV infected pigs	[131,136]		
*PCV2*	Impairment of pDCs ability to produce IFN-α through TLR-7 and TLR-9 receptors-mediated pathway	[188]	Induction of IFN-γ following vaccination with inactivated PCV2 vaccine and VLPs-based vaccine	[174,195]
	In vivoidetectionof IFN-α response in PCV2 infected piglets and pigs	[189,191]	Failure in inducing an IFN-γ response after vaccination with Non-Assembled ORF2 Capsid Protein of Porcine Circovirus 2b	[196]
	Detection of high levels of IFN-α in PCV2-infected swine alveolar macrophages (AMs)	[192]		
	Induction of IFN-β in PK-15 cells through a RIG-1 and MDA-5 signaling pathway	[193]		
*PRV*	Induction of IFN-α response in purified pDCs	[224]	Induction of a persistent IFN-γ response in PBMCs following PRV infection	[229]
	increased IFN-α response associated with PRV gE-gI-deleted mutants	[225]	In vitro induction of IFN-γ response in whole blood saples from PRV-vaccinated pigs	[230]
	Down-regulation of IFN-β production by PRV gE via CREB-CBP degradation	[226]		
*ASFV*	In vitro up-regulation of IFN-α subtypes in unactivated and activated moMφ infected with both virulent and attenuated strains	[265]	Suppression of type II IFN response following infection with ASFV-encoded multigene families (MGFs) strains	[270,271]
	In vitro statistically significant up-regulation of IFN-α10, IFN-α12, IFN-α13, IFN-α15, IFN-α16, IFN-α17 and IFN-β in unactivated moMφ infected with an attenuated strain	[265]	In vivo increased levels of IFN-γ following infection with ASFV virulent strain	[272]
	In vitro statistically significant up-regulation of IFN-α1, IFN-α10, IFN-α15, IFN-α16 and IFN-α17 in activated moMφ infected with an attenuated strain	[265]		
	Up-regulation of IFN-β expression in infected Mφ with attenuated strain through cGAS-STING-IRF3 signaling pathway during early infection	[248]		
	Suppression of type I IFN response following infection with ASFV-encoded multigene families (MGFs) strains	[270,271]		
	In vitroinduction of high levels of IFN-β mRNA after infection with deleted-MGF360 and MGF530/505 Benin 97/1 strain	[250]		
	In vivo increased levels of type I IFN following infection with ASFV virulent strain	[259,272,273]

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
