# Peer review of "The Swine IFN System in Viral Infections: Major Advances and Translational Prospects"

_pathogens, 2022, doi:10.3390/pathogens11020175_

Round 1

Reviewer 1 Report

This manuscript (Razzuoli et al.) is an excellent comprehensive review focusing on the major viral diseases of swine which animal is of major importance for the farming sector. This is a very well written well-structured manuscript. The review describes the three RNA and also three DNA viruses of swine, describes their associated diseases and discusses the animal cellular defense mechanisms during viral infection and the viral mechanisms of counteracting this innate immune response. Furthermore, the major strength of the manuscript is the likelihood of targeting broader interest readers with diverse background due to the timely in-depth discussion of mechanism, diseases and also translational studies on these viruses. The major weakness observed is that the dense information content is not balanced with any summary table or illustration to help understanding the material. 

Specific points of the reviewer:

  • Authors should include at least one comprehensive figure highlighting their described mechanisms. They should improve the manuscript by including a larger figure , for example by representing the major events particularly for the three distinct classes of IFN and the responses, and show how various viruses affect these pathways, as relevant.
  • The manuscript has a sufficient number of literature and has proper citation for relevant statements, however they should avoid referring to unspecific statements to previous sections (#615: ”PRRS cases referred to in a previous chapter.” ; or #636 “in a previous chapter”; or #150 “by the aforementioned..”) Refer more clearly, and include the chapter number as the reference point.
  • #71: The sentence refers to “SOC3” [protein] name should be corrected to “SOCS3”.
  • #81: Unclear transition “In this conceptual..” Elaborate more, or re-phrase this next sentence, to make it clear how this connects to the previous statement.
  • #473: If this is a title, it should be more clear in formatting or edited.
  • #534 and #92: The sections are starting with the exact same half sentences, consider changing them to make them more individualized.
  • Review sections are generally well written, but some of the sections end abruptly, without summarizing the major take home message. Example include #589, #621.
  • #772: Explanation is missing on the importance of CREB-binding protein in this context. Explain the protein importance here or remove.
  • #804: “kpb” should be corrected to “kbp” (if it refers to kilobase pair).

Author Response

Thank you for providing us the opportunity to revise and improve the manuscript following the constructive critique provided by the reviewers. In response to the two reviewers comment the text, were modified. Below, we provided a point-by-point response (Please see the attachment), in italics, to each of the reviewers' comment and enclose the manuscript with the corresponding revisions shown as red text.

Reviewer 2 Report

Except for minor spelling mistakes this review is ready for publication. 

Current review gives an excellent overview of interferons (specific type of cytokines) and their (in)actions in pigs during RNA and DNA viruses infections. 

Author Response

Thank you for providing us the opportunity to revise and improve the manuscript which has been re-edited by all authors